# Investigating the practices and preferences of health scholars in sharing open research data

**Muhamad Prabu Wibowo** [1,2]*, **Lorri Mon** [2]

1 Department of Library and Information Science, University of Indonesia, Depok, Jawa Barat, Indonesia,
2 School of Information, Florida State University, Tallahassee, Florida, United States of America

* mprabuw@ui.ac.id

## Abstract

This survey-based study examines health science scholars' perceptions of collaborative research behavior and sharing open research data in university settings. A total of 362 health science scholars from U.S. universities participated in an online questionnaire consisting of 59 questions. Descriptive and inferential statistical analyses of the data included frequencies, cross-tabulations, descriptive ratio statistics, and the non-parametric Kruskal-Wallis one-way analysis of variance. Four open-ended questions were also analyzed to provide further insights into the survey findings. The study reveals that health scholars share their data with colleagues within their institution or project, demonstrating a lesser inclination toward open research data sharing practices through institutional repositories and journal supplements. Motivating factors and challenges influencing researchers' decisions to share their research data were also identified. While scientific and knowledge advancement served as major incentives, health scholars working with human-related data expressed concerns about privacy and confidentiality breaches, which are primary barriers to data sharing. Some participants indicated that requirements and policies also influenced their willingness to share data. Disciplinary variations were observed regarding data-sharing practices through journal supplements, secondary data analysis, and personal communication. Furthermore, significant differences emerged between funded and non-funded scholars, impacting their practices, motivations, and challenges in sharing open research data. Important factors driving health science scholars to share open research data include resources, policy compliance, and requirements. This study contributes valuable insights for policy development by investigating factors that can foster openness and sharing of research data in the health sciences. The findings shed light on the complexities and considerations associated with open data-sharing practices, enabling stakeholders to develop effective strategies and frameworks.

## Introduction

The increasing trend of sharing open health sciences data helps shape e-science, modern society, and industry—allowing others to utilize widely available health data for knowledge

**Data Availability Statement:** The data are available on Figshare. The DOI for accessing the dataset is 10.6084/m9.figshare.24187362.

**Funding:** The study is funded by the American Indonesian Cultural and Educational Foundation

(AICEF) dissertation travel grant and the Florida State University, School of Information, Esther Maglathlin Doctoral Research Scholarship. The funders had no role in study design, data collection and analysis, decision to publish, or preparation of the manuscript.

**Competing interests:** The authors have declared that no competing interests exist.

development and better public health services. Sharing open health data contributes to science and knowledge development and helps transform modern society. Open data on hospital beds was used to facilitate emergency response during Hurricane Irene [1]. Open health data is also used in university education curricula, such as for data analytics courses. Health data obtained from wearable sensors is also used for predictive machine learning [2]. Sharing open health data can induce collaboration that brings innovations of new treatments and new knowledge about the causes of diseases and informs personalized individual care [3]. Publicly available health data brings many potential benefits: supporting state and community-level health planning, assisting emergency response, improving efficiency, fueling innovation, empowering consumers to make informed decisions, and improving public health practices [4]. Open data in healthcare also contributes to strengthening public health surveillance [5]. Open health data can improve healthcare delivery and policy [6]. It improves early detection of health and environmental threats due to the nature of open data that enhances collaboration and administration. Shared and open health data provide opportunities to improve health systems and individual care [3]. Open health data has great potential to contribute to the knowledge of disease, increasing the quality of diagnostics and delivery of healthcare and treatment. It fuels innovation that allows for the development of new products and services [7].

Despite the many potential benefits, sharing open health data also brings concerns, issues, and challenges. A previous study shows that awareness and utilization of publicly available health data utilization is low in university environments [4]. In university environments, there also is a disconnect between the academic reward system and open data sharing practices in university environments, making researchers unwilling to share their data in open repositories [8]. According to Nosek et al. [8], there are complexities in coordination of open data sharing between universities, granting agencies, and publishers. Another problem is a lack of studies on integrating open data with policies, licenses, technology, financing, organization, culture, and legal frameworks [9, 10]. There is a conflict between making research data publicly available and commercializing research results [11]. Furthermore, potential privacy breaches, copyright infringement, and confusion over data ownership hinder the implementation of open and shareable health sciences research data [3]. Technical aspects, such as high cost can also hinder health data sharing [12]. In emergencies such as pandemics where health data sharing is needed, mechanisms for sharing research data are often lacking [13].

This study aims to gain insights into the factors influencing health scholars' willingness to share open research data. These insights can guide the development of strategies for addressing the challenges and issues in sharing health research data, such as improving awareness, infrastructure, coordination, and legal frameworks, while minimizing issues and challenges, such as ensuring data privacy and ownership.

While previous studies have explored factors influencing sharing open research data in the broader landscape of research data management [14–17], this study addresses the remaining gap of understanding perceptions and tendencies of scholars in the health sciences field towards sharing health research data. Sharing health data is gaining importance nowadays, especially since the recent pandemic [18], and this study can enhance a collaborative culture by identifying practices for and sharing open health research data.

This study seeks to answer the following research questions:

RQ1: What are the preferences of health scholars regarding sharing their research data?

RQ2: Is there a significant difference in the attitudes towards sharing open research data among health scholars from different disciplines?

RQ3: Is there a significant difference in the attitudes towards sharing open research data between funded and non-funded health scholars?

RQ4: What are the important factors that influence health scholars to share open research data?

RQ5: How do health scholars perceive issues and challenges in open research data and data sharing?

It is essential to examine health science scholars' unique perceptions, awareness, and tendencies to promote openness, collaboration, and sharing effectively.

## Theoretical and conceptual framework

### Practices of sharing open research data

Research data can be shared in various ways. Collaborative work is closely related to research data sharing [19]. Collaborative work, including research collaboration and data sharing, can be developed according to collaborators' geographic, cognitive, organizational, institutional, and social proximity [20–24]. Sharing research data is also a part of the open science movement and an effort to increase transparency benefits science and society [25–27]. Sharing health research data can be done through open data repositories, journal supplements, and peer-to-peer sharing [25, 28].

In a broader sense, research data sharing is an essential part of open research data, as it ensures that datasets are made publicly accessible for wider use. Open research data represents a strategic approach aligned with data management guidelines, designed to make data available for public reuse and redistribution [29, 30]. By prioritizing data transparency, open research data initiatives standardize data formats and develop platforms that facilitate easy access to shared datasets. Such practices foster data sharing and collaborative efforts, which are critical in today's networked environment and modern scholarly communication [28, 31]. The open data initiative further expands the availability of data on a global scale, promoting widespread data sharing [32, 33]. Many grant agencies worldwide require researchers who receive funding to deposit their data into open research data repositories [34–36]. Open research data is often a mandatory condition for publicly funded research [28, 37]. Since publicly funded data is considered public property, researchers are obligated to make it freely accessible, complying with policies and requirements set by funding bodies.

In health sciences, a vast range of clinical and healthcare data can be made publicly accessible as part of open research data initiatives [7]. Many researchers are opening their administrative and operation data and health-related survey [4]. Opening health research are critical and relevant in today's scientific research in enhancing collaborative culture to solve various problems faster, such as finding a cure [18].

There is a set of principles, norms, and rules to be followed when making research data open and shareable. Data should be discoverable, modifiable, non-proprietary, and reusable as permitted by law [4]. Before data can be shared, it should be prepared and assessed carefully to minimize the risk of privacy breaches through deidentification and other sensitive data protection measures [7]. Any human subject research data should meet the criteria of an ethical data sharing policy before being made available to the public [38]. Many potential types of health data can be shared such as administrative health data, clinical data, health-related surveys, surveillance data, and social media data [4]. Some open data repositories also have established rules on which data formats can be shared, including image research data [39].

### Driving factors for sharing research data practices

Many previous studies explored factors influencing sharing of open research data [14–17, 40]. Zuiderwijk et al. [14] conducted a literature review and provided a comprehensive list of driving and inhibiting factors that may influence researchers to practice open data sharing. These factors are divided into eight main categories: researcher's background, requirements and formal obligations, personal drivers/intrinsic motivations, facilitating conditions, trust, expected performance, social influence and affiliation, effort, researcher's experience, legislation and regulation, and data characteristics.

Kim and Adler [40] tested individual motivations, institutional pressures, and availability of resources in facilitating scientists' data sharing. According to their survey, data-sharing behaviors are driven by personal motivation, including perceived career benefits and risks, perceived efforts, and attitude toward data sharing. Another important factor is perceived normative pressure. The other potential factors influencing data-sharing behaviors are funding agency pressure, journal pressure, and the availability of data repositories. Sayogo and Pardo [16] conducted a survey study investigating challenges in sharing research data. They identified key factors influencing researchers to share data: data management skills, organizational support, and legal and policy requirements. Zhu [15] conducted a survey study to factors that may influence researchers in sharing research data, including gender, age, seniority, discipline, awareness of social media, awareness and attitudes toward sharing data, and open access publishing policy. Zhu's study suggests significant differences in gender, age, and seniority influence research data-sharing practices. According to Zhu's survey, social media can increase readership and citation. Mozersky et al. [17] surveyed qualitative researchers on factors that made them likely to share their data. The study revealed that researchers were more likely to share their data when doing so would increase the societal impact of their research and facilitate increased collaboration. They also found that several disciplines are more inclined to qualitative data sharing.

Open research data sharing can be driven by the perceived benefits researchers can get, such as advancement of knowledge and science development, society development, collaboration, co-authorship, and transparency [8, 14, 33, 41, 42]. Open research data sharing can be encouraged with incentives, data citation practice, and interaction between data repositories and journals [43]. Open data facilitates communication with other researchers [26]. Many researchers are willing share their data if specific conditions are met, such as awards and recognition [44].

Sharing open research data can be influenced by external factors, such as institutional and national policies [11, 34–36, 45]. According to several papers, open research data is driven by policies, pressure, and grant requirements [16, 25, 28, 30, 34, 37, 40, 45]. Kostkova et al. [3] discussed the influence of data ownership and national regulations on the practice of open research data in health sciences.

### Challenges and issues in sharing research data

Despite the potential benefits, many scholars are not engaged in open data sharing practices for many reasons. Early studies showed that openness is often avoided, such as in the military, non-disclosure of information to protect competitive advantages, such as trade secrets [46, 47]. Open data sharing brought licensing and ownership issues in many U.K. e-science projects [27]. At the beginning of the open data policy implementation of the Bermuda Principles in 1996, disputes arose because of data ownership boundaries [48].

Sharing open research data raises many issues and challenges. Open research data are prone to potential misuse, such as data that can be taken out of context [49, 50]. Insufficient

knowledge of the context of how the data is being used may result in misinterpretation or mis-application of research data [50, 51]. This has resulted in researchers fearing the exploitation or inappropriate use of their data [11]. Mozersky et al. [17] identified concerns about sharing qualitative research data, including losing control over the data, financial costs, technological failures, and lack of knowledge.

Sharing open research data has brought many issues in human subjects research, where there are perceived risks around confidentiality, anonymity, data security, and study sensitivity [52]. The withholding of data can be traced back to technological, social, organizational, eco-nomic, legal and policy, and behavioral aspects [16]. Some researchers declined to share their data because of concerns regarding the privacy and confidentiality of data, a desire to maxi-mize the professional and economic benefits, and perceiving data sharing as a threat to their intellectual property [53]. Many scholars delay sharing their research data due to waiting for the publication of the research [54]. Avoidance of open research data sharing can also be related to several individual-level barriers: wishes to retain exclusive use of the data, lack of career rewards for sharing, lack of time and resources, lack of experience and expertise in data management and sharing, and fear of exploitation or inappropriate use of data [8, 11, 29, 49, 55, 56].

Besides identifying driving factors, Zuiderwijk et al. [14] identified a comprehensive list of factors that inhibited sharing open research data that are grouped into several categories, including researchers' backgrounds (e.g. low level of involvement and low experience), requirements and obligations (e.g. lack of requirement), personal drivers (e.g. laziness or fear of reuse), facilitating conditions (e.g. lack of repository, platform or budget), trust (e.g. issues of trust and potential harm), and expected performance (e.g. reduced expected performance). Many researchers avoid sharing research data due to a lack of time, skills, facilities, and infra-structures [8, 11, 44, 49, 56, 57]. Other challenges are related to conflicts with epistemological and methodological principles, and ethical concerns [57].

There are many indirect risks of open research data. The increased transparency of open research data can undermine privacy, confidentiality, and anonymity, and pose further chal-lenges of disclosing potentially sensitive information, thus increasing the risk of privacy breaches and security violations [49, 55, 58]. Privacy and security issues can also result from researchers' lack of awareness of proper open research data practices or guidelines [49].

## Methods

This study aims to understand factors that influence health scholars to get involved in sharing open health research data. This paper uses a survey design to evaluate health scholars' aware-ness of the importance of policies, practices, driving factors, issues and challenges of sharing open research data. An online questionnaire using Qualtrics was designed that includes four constructs: demographic questions, open research data practices, driving factors and triggers in open research data, and challenges and issues in open research data.

The first construct is related to demographics, such as level of education, involvement in research activities, and research experiences. The demographic construct also covers questions about participants' involvement in research grants, research collaboration, and previous expe-riences in secondary data analysis. According to several studies, demographic factors may influence researchers' practices in research data sharing [14, 15, 17, 19].

The second construct consists of items to accommodate responses about ways of sharing open research data, such as how willingly researchers share their research data, and their understanding of what kinds of data can be made open to the public. Research data can be shared in various ways, through data repositories, journals, peer-to-peer, and proximity [20–

25, 28, 34]. There are also deidentification process in making research data open and available to the public [7, 38].

The third construct consists of items related to driving factors and triggers in open research data including benefits gained by sharing research data [8, 11, 14, 25–27, 29, 33, 41, 42, 45, 51, 59]. National grants and institutional policies can also motivate researchers' willingness to comply with policies and laws [3, 16, 25, 28, 30, 34, 37, 40, 45].

The fourth and last construct tries to understand how health science scholars perceive the challenges, issues, risks, and barriers in open research data, such as lack of resources, lack of time, competence, risks of privacy, and confidentiality breaches [27, 29, 46–48, 50, 54, 55, 57, 58, 60].

Most measurement items accommodate five-point Likert scales asking the degree of agreement, anchored between strongly disagree (1) and strongly agree (5). There are also questions representing items with multiple-choice answers. The questionnaire draft was pre-tested with 12 participants to ensure that the questions were easy to understand and clear, and that the questionnaire could be answered within the planned 15 minutes. The final version of the questionnaire consisted of 59 questions.

This study employed a non-probability sampling method since identifying and accessing all health sciences scholars in the U.S. is challenging. The focus of this study on the U.S. was justified by the country's maturity in practicing open and sharing health research data. The U.S. is recognized as one of the leading countries in providing open data and data repositories, with numerous government institutions and universities hosting health data repositories. In 2019, the U.S. was one of the countries providing the highest number of open data repositories [61]. Many U.S. government institutions host open health data, such as the National Institute of Health (NIH) [62, 63]. It is also very common for U.S. universities to host data repositories [64]. Therefore, the U.S. was selected as the research setting to capture best practices in open and sharing health research data.

The list of participants was gathered from public domain databases and websites, such as the Scopus database. The Scopus database allows filtering of searches through specific fields, such as publication years. The questionnaire was sent to approximately 11,636 potential participants from October 1st, 2022, to January 31st, 2023. The consent was written and provided to the participants at the beginning of the questionnaire before they proceeded. This study was exempted by the FSU ethics committee board under the study ID 'STUDY00003055.' There were 1,644 or 14.1% undelivered emails because recipients were inactive or no longer in the department. The survey gathered 487 responses. After further checking the responses, there were 362 usable valid responses or a 3.1% response rate. The survey data for this study has been made available online at Figshare [65].

The survey responses were analyzed using SPSS 26 and a spreadsheet to examine the constructs and variables. Descriptive analysis techniques were predominantly employed, including frequencies, cross-tabulations, and charts. A one-way ANOVA was planned to ascertain the average differences among health scholars from different groups of disciplines and grant funding. Upon conducting post-hoc testing, it was found that the assumption of homogeneity of variances was violated, making the parametric ANOVA unsuitable. Additionally, since the dependent variable in this study is ordinal, a non-parametric alternative, the Kruskal-Wallis test, was deemed more appropriate. The Kruskal-Wallis test does not assume normal distribution or equal variances across groups, making it ideal for comparing mean ranks between three or more independent groups when the assumptions for parametric tests are not met [66, 67]. As a result, this method was employed to identify significant differences in the sharing of research data across various disciplines and funding sources. Additionally, four open-ended questions were included in the survey, allowing respondents to provide text responses. These

open-ended questions were analyzed using thematic analysis by finding patterns to provide further insights into the survey findings.

## Findings and discussions

### Demographic information of health sciences scholars

This section presents the findings and analysis of a survey, starting with an overview of the participants' demographic information, which is presented in Table 1.

Regarding research disciplines, most participants come from medical science (34,3%), followed by public health at 18%, and nursing at 11.3%. Health informatics and information technology, psychology, and veterinary science represented 8.8%, 5.8%, and 5% of participants, respectively. Pharmaceutical science and healthcare administration represented 4.7% and 4.4% of participants, respectively. Dentistry had the smallest percentage at 2.2%, and 5.5% of participants chose the "Other" category. Other category discipline consists of scholars in nutrition, educational health, social work, and many more.

Regarding the level of education, among the 362 respondents, the majority hold a doctoral degree, comprising 59.4% of the total. Professional degree holders, such as JD, MD, and DVM, represent 30.9%. Meanwhile, those who have completed a Master's degree make up 8% of the participants. A small percentage of respondents have a Bachelor's degree in college (1.1%), an Associate degree (0.3%), and some high school or college but no degree (0.3%). Most participants had 21 years of experience and above, accounting for 30.7% of the total sample. The second largest group was those with 6–10 years of experience (18.2%), followed closely by those with 11–15 years of experience (18%) and those with 16–20 years of experience (17.4%). The smallest group was those with 1–5 years of experience, accounting for only 15.7% of the total sample. The survey also gathered data on the number of research collaborations that participants engaged in within a year. Of 362 respondents, 21.3% reported having less than three research collaborations within a year, while 40.1% reported having 3–5 collaborations. Meanwhile, 25.4% reported having 6–10 collaborations, and only 7.7% reported having 11–15 collaborations. The smallest percentage of respondents, at 5.6%, reported having more than 15 research collaborations within a year.

The survey asked participants how often they have reused publicly available research data as a secondary data study in their career as a scholar. The majority of the participants (33.1%) reported never reusing publicly available data. This was followed by 21.8% who have reused data 1–2 times. Meanwhile, 16.3% reported reusing data 3–5 times, 11.3% reported reusing data 6–10 times, and only a small percentage reported reusing data 11–20 times (5.5%) or more than 20 times (6.4%). A few participants (5.5%) did not disclose or were missing information about using others' publicly available data. The finding still confirms Martin et al.'s [4] study that reported public health research data utilization is low in a university environment.

One of the rationales for the low reuse of publicly available data was the lack of context, as expressed by a participant:

"I wouldn't use this same type of data collected by other people that I don't know—it wouldn't be as rich to me without the context/verifying how it was created. I would rather take their tools/methods and do it myself"

(#226, healthcare administration, 6–10 years of experience)

Regarding the funding, out of the 362 respondents, 85 (23.5%) of participants' research was unfunded or self-funded. The majority, 182 (50.3%), were funded nationally, while 91 (25.1%) were institutionally funded. There were four (1.1%) missing responses. Of the 362

**Table 1. Demographic information.**

| Category | Demographic information | Frequency | Percentage |
|---|---|---|---|
| Research Discipline | Medical Science | 124 | 34.3 |
| | Public Health | 65 | 18 |
| | Nursing | 41 | 11.3 |
| | Health Informatics and Information Technology | 32 | 8.8 |
| | Psychology (Mental Health) | 21 | 5.8 |
| | Veterinary Science | 18 | 5 |
| | Pharmaceutical Science | 17 | 4.7 |
| | Healthcare Administration | 16 | 4.4 |
| | Dentistry | 8 | 2.2 |
| | Other | 19 | 5.5 |
| | Total | 362 | 100 |
| Level of Education | Some high school or college but no degree | 1 | 0.3 |
| | Associate degree in college (2-year) | 1 | 0.3 |
| | Bachelor's degree in college (4-year) | 4 | 1.1 |
| | Master's degree | 29 | 8 |
| | Doctoral degree | 215 | 59.4 |
| | Professional degree (JD, MD, DVM) | 112 | 30.9 |
| | Total | 362 | 100 |
| Research Experience | Less than five years | 57 | 15.7 |
| | 6–10 years | 66 | 18.2 |
| | 11–15 years | 65 | 18 |
| | 16–20 years | 63 | 17.4 |
| | 21 years and above | 111 | 30.7 |
| | Total | 362 | 100 |
| Number of Collaborations within a year | Less than three research collaborations | 77 | 21.3 |
| | 3–5 research collaborations | 145 | 40.1 |
| | 6–10 research collaborations | 92 | 25.4 |
| | 11–15 research collaborations | 28 | 7.7 |
| | More than 15 research collaborations | 20 | 5.5 |
| | Total | 362 | 100 |
| Number of times reusing others' publicly available data | Never | 120 | 33.1 |
| | 1–2 times | 79 | 21.8 |
| | 3–5 times | 59 | 16.3 |
| | 6–10 times | 41 | 11.3 |
| | 11–20 times | 20 | 5.5 |
| | More than 20 times | 23 | 6.4 |
| | *Missing/Not disclosing* | 20 | 5.5 |
| | Total | 362 | 100 |
| Funding of Collaborative Projects | Unfunded/Self-funded | 85 | 23.5 |
| | Institutionally funded | 91 | 25.1 |
| | Nationally funded | 182 | 50.3 |
| | *Missing/Not disclosing* | 4 | 1.1 |
| | Total | 362 | 100 |

(*Continued*)

**Table 1.** (Continued)

| Category | Demographic information | Frequency | Percentage |
|---|---|---|---|
| Value of winning grants within the last five years | Not receiving any grant | 88 | 24.3 |
| | Less than USD10,000 | 23 | 6.4 |
| | USD 10,001–50,000 | 24 | 6.6 |
| | USD 50,001–100,000 | 19 | 5.2 |
| | USD 100,001–500,000 | 47 | 13 |
| | USD 500,001–1,000,000 | 34 | 9.4 |
| | More than USD 1,000,000 | 100 | 27.6 |
| | *Missing/Not disclosing* | 27 | 7.5 |
| | Total | 362 | 100 |

respondents, 89 reported not receiving any grant during this period. Among those who did receive grants, 23 reported receiving less than USD $10,000, while 24 reported receiving between USD $10,001–50,000. Nineteen respondents reported winning grants between USD $50,001–100,000, while 46 reported winning grants between USD $100,001–500,000. Thirty-four respondents reported winning grants between USD $500,001–1,000,000. Additionally, 100 respondents reported winning over USD $1,000,000 in grants over the past five years. However, 27 respondents did not disclose the value of grants received.

## Practices of sharing open research data among health scholars

This section explains practices of sharing research data by health scholars, shown in Table 2.

Table 2 presents the survey results on practices in sharing research data among health science scholars. Results indicate that, on average, participants share their research data with other members in the same research project, with a mean score of 4.74, ranked first among all the statements. This suggests that health scholars are more willing to share their data with their immediate colleagues within the same project. Participants also reported a high level of sharing their health research data with scholars in the same institution, with a mean score of 4.02, ranked second. This implies that health scholars are more willing to share their data within their institutions. The ranking was followed by sharing through informal ways (3.68); sharing outside institutions (3.57); opening the research data that are funded by grants (3.53); conducting secondary data analysis on publicly available data (3.29); open research data through journal supplement (3.25); and open research data through institutional repositories (2.87). Results suggest that surveyed health scholars prefer to share their data through less formal and smaller

**Table 2. Health scholars' practices of sharing health research data.**

| Rank | The practice of Sharing Health Research Data | Mean | SD |
|---|---|---|---|
| 1 | I share my health research data with other members in the same research project. | 4.74 | 0.618 |
| 2 | I share my health research data with researchers in the same institution. | 4.02 | 1.192 |
| 3 | I share my health research data in informal ways (personal communication). | 3.68 | 1.461 |
| 4 | I share my health research data with researchers outside of my institution. | 3.57 | 1.339 |
| 5 | I open my health research data to public for studies that are funded by grants. | 3.53 | 1.390 |
| 6 | I conduct secondary data analysis through health sciences data that are available to public | 3.29 | 1.608 |
| 7 | I open my health research data to public through journal supplement | 3.25 | 1.475 |
| 8 | I open my health research data to public through repositories in my institution | 2.87 | 1.470 |

**Table 3. Test for group differences across disciplines of health scholars on practices of sharing research data.**

| The practice of Sharing Health Research Data | Kruskal-Wallis H | df | Asymp. Sig. |
|---|---|---|---|
| I share my research data with other members in the same research project. | 2.205 | 9 | 0.988 |
| I share my health research data with researchers in the same institution. | 8.646 | 9 | 0.471 |
| I share my research data in informal ways (personal communication). | 19.38 | 9 | 0.022 * |
| I share my health research data with researchers outside of my institution. | 12.836 | 9 | 0.17 |
| I open my research data to public for studies that are funded by grants. | 9.515 | 9 | 0.391 |
| I conduct secondary data analysis through health sciences data that are available to public | 42.191 | 9 | 0.000 ** |
| I open my research data to public through repositories in my institution | 12.676 | 9 | 0.178 |
| I open my research data to public through journal supplement | 25.764 | 9 | 0.002 ** |

\* p < 0.05;

\*\* p < 0.01.

scope data-sharing channels. They prefer internal and personal methods of sharing health research data instead of through formal repositories and public platforms.

The study then compared the practice of sharing research data according to the group of disciplines of health scholars using the Kruskal-Wallis test. According to the Kruskal-Wallis test, significant differences were observed among disciplines regarding their research data sharing practices. These differences were found in three specific areas: sharing research data in informal ways (personal communication) (H = 19.38, df = 9, p < 0.05), conducting secondary data analysis through publicly available health data (H = 42.191, df = 9, p < 0.01), and open research data through journal supplements (H = 25.764, df = 9, p < 0.01), as presented in Table 3.

The mean ranks for sharing research data in informal ways (personal communication) according to disciplines are as follows (ascending): public health (MR = 142.38), healthcare administration (MR = 165.44), nursing (MR = 167.6), psychology (MR = 175.67), health informatics and information technology (MR = 176.25), other (MR = 178.43), veterinary science (MR = 185.69), pharmaceutical science (MR = 201.12), medical science (MR = 202.17), dentistry (MR = 228.94). According to pairwise comparisons, public health has the lowest mean rank, significantly different from medical science, pharmaceutical science, and dentistry.

The mean ranks for conducting secondary data analysis through health research data that are available to the public, according to disciplines, are as follows (ascending): veterinary science (MR = 102.24), psychology (MR = 127.74), nursing (MR = 144.69), medical science (MR = 172.75), pharmaceutical science (MR = 174.32), other (MR = 188.11), dentistry (MR = 199.31), health informatics and information technology (MR = 202.23), public health (MR = 221.95), and healthcare administration (MR = 247.5). According to pairwise comparisons, veterinary science, psychology, and nursing have low mean ranks that are significantly different from other disciplines.

The mean ranks for opening research data through journal supplements, according to disciplines, are as follows (ascending): psychology (MR = 122.45), nursing (MR = 139.8), healthcare administration (MR = 162.57), public health (MR = 164.95), medical science (MR = 190.46), health informatics and information technology (MR = 197.03), pharmaceutical science (MR = 208.59), other (MR = 209.53), veterinary science (MR = 216.25), and dentistry (MR = 244.44). According to pairwise comparisons, psychology and nursing have low mean

**Table 4. Test for group differences across grant funding of health scholars on practices of sharing research data.**

| The practice of Sharing Health Research Data | Kruskal-Wallis H | df | Asymp. Sig. |
|---|---|---|---|
| I share my research data with other members in the same research project. | 2.976 | 2 | 0.226 |
| I share my health research data with researchers in the same institution. | 1.611 | 2 | 0.447 |
| I share my research data in informal ways (personal communication). | 6.62 | 2 | 0.037 * |
| I share my health research data with researchers outside of my institution. | 1.367 | 2 | 0.505 |
| I open my research data to public for studies that are funded by grants. | 4.792 | 2 | 0.091 |
| I conduct secondary data analysis through health sciences data that are available to public | 1.532 | 2 | 0.465 |
| I open my research data to public through journal supplement | 0.714 | 2 | 0.7 |
| I open my research data to public through repositories in my institution | 1.546 | 2 | 0.462 |

\* $p < 0.05$;

\*\* $p < 0.01$.

ranks that are significantly different from other disciplines. The comparisons also show significant differences between public health and dentistry.

The Kruskal-Wallis test was performed on the group of grant funding. It revealed statistically significant differences between funding on sharing research data in informal ways (personal communication) ($H = 6.62$, $df = 2$, $p < 0.05$), presented in Table 4.

The mean ranks for sharing research data in informal ways (personal communication) based on funding are as follows (ascending): institutionally funded (MR = 158.91), nationally funded (MR = 179.63), and unfunded/self-funded (MR = 197.05). According to pairwise comparison, unfunded/self-funded health scholars have a significantly higher mean rank in sharing research data through informal ways than funded scholars.

According to the analysis of open-ended questions, several participants expressed that the practice of sharing health research data vary from one project to another and depends on many things, including journal requirements and grant funding. The answers explain the variances in health scholars' practices of sharing research data.

"Different journals and funders have different requirements for data sharing. I generally follow the minimum data sharing required"

(#141, medical sciences, 5–10 years of experience)

"Each project is different."

(#352, medical sciences, more than 20 years of experience).

"..different projects have different needs"

(#370, public health, 10–15 years of experience)

"It varies widely & depends on the DUAs on a project by project basis"

(#436, health statistics, 21 years and above of experience)

Several participants also expressed that sharing health research data in industry-business-for-profit is different in non-profit organizations such as university research health data settings.

"In industry, much research data that are adverse (i.e., related to a potential or no adverse impact on health) are not published. Much industry-sponsored research is not open to the public"

(#310, pharmaceutical science, 21 years and above of experience)

"Several projects are industrially funded or tied to protected IP so we cannot share data"

(#331, pharmaceutical science, 16–20 years of experience)

"our projects are mainly funded by industry. Aggregate deidentified data queries are a major source of revenue for our organization. . . different with traditional academic universities and hospitals.. maintaining proprietary data is the way that non-profits can fund research and free programs for people living with illness."

(#437, clinical psychology, 16–20 years of experience)

"I receive funding mostly from the pharmaceutical industry so it's often that I don't have ownership of the data nor is it public."

(#418, pharmaceutical science, 16–20 years of experience)

Regarding the differences among projects, funding sources, and the nature of data, health scholars have expressed that sharing research data requires understanding policies, agreements, and laws. These policies, agreements, and laws serve as guidelines for scholars when sharing research data, and they can vary from one subject to another and from one institution to another.

"Data sharing in often governed by data sharing agreements"

(#152, healthcare administration, 11–15 years of experience)

"We are limited by our institution policy"

(#212, public health, 6–10 years of experience)

"Depends on IRB approval, institution policies"

(#237, nursing, 11–15 years of experience)

"My institution and we have a policy / process for sharing"

(#243, medical science, 16–20 years of experience)

"HIPAA governs a lot of what I can do with my data!"

(#252, health informatics and information technology, 16–20 years of experience)

"Depends on IRB approval and data ownership

(#256, public health, 11–15 years of experience)

"My workplace has several institutional barriers to sharing information."

#267

"Dependent on privacy laws governing PHI and HIPAA and the institutional policies

(#346, medical science, 21 years and above of experience)

"Sharing health data is always guided by IRB guidelines."

**Table 5. Health research data type and format being shared.**

| Category | Item | Frequency | Percentage |
|---|---|---|---|
| Research Data Type | Observational data | 226 | 62.4 |
| | Surveys and statistical data | 217 | 59.9 |
| | Experimental | 152 | 42 |
| | Data models | 71 | 19.6 |
| | Simulation data | 32 | 8.8 |
| | Reference or canonical data | 11 | 3.0 |
| | Remote-sensed data | 9 | 2.5 |
| | Other | 58 | 16 |
| Research Data Formats | Numerical | 279 | 77.1 |
| | Text | 209 | 57.7 |
| | Spreadsheet | 179 | 49.4 |
| | Images | 81 | 22.4 |
| | Video | 28 | 7.7 |
| | MATLAB | 11 | 3 |
| | Other | 27 | 7.5 |

(#360, medical science, 21 years and above of experience)

"Laws related to animal research are substantially different"

(#377, veterinary sciences, 16–20 years of experience)

The study also aims to investigate data types and formats of research data being shared by health scholars, as presented in Table 5.

Based on the data presented in Table 5, the most common research data type used by participants is observational data, with 62.4% of respondents reporting its use. Surveys and statistical data are also commonly used, with 59.9% of respondents reporting their use. Experimental data are reported by 42% of respondents, while data models and simulation data are used by 19.6% and 8.8% of respondents, respectively. The least commonly used research data types are reference or canonical and remote-sensed data, with only 3% and 2.5% of respondents reporting their use, respectively. Additionally, 16% of respondents reported using other types of research data. The other research data types mentioned by respondents are health records, registry data, legal documents, interviews, neuroimaging, and clinical trials.

Table 5 also presents research data formats being used by health scholars in the U.S. Most respondents were working with numerical format (77.1%), followed by text (57.7%) and spreadsheet (49.4%) formats. Images and videos were also used but to a lesser extent (22.4% and 7.7%, respectively). Some respondents used MATLAB (3%). 7.5% of respondents used other data formats, including pdf, DNA, graphs, audio recording, claims, geospatial, SAS, and Stata.

According to several participants, there is no clear preference for sharing specific types of research data. In some cases, survey data may be shared, while in other cases, it may not be shared. The ability to share research data is influenced by various factors, including the level of data aggregation, the specifications outlined in the consent form regarding data sharing, and the feasibility of deidentifying the data.

"Some "health-related survey data" can be made open to the public and others cannot, depending on other factors."

(#141, medical science, 6–10 years of experience)

"Sharing data depends largely on the type of data we have, participant privacy, and participant consent."

(#227, health informatics and information technology, 6–10 years of experience)

"It depends: (1) if the data is aggregated (more likely to be shared) or disaggregated; (2) how the consent form states what data will be shared; (3) data cannot be deidentified

depends on the consent received, nature of the data, and whether or not it can be directly or indirectly connected to a person or population."

(#248, health informatics and information technology, 11–15 years of experience)

From open-ended questions, participants expressed that most deidentified and anonymized data can be shared as long as compliance is maintained with agreements, policies, laws, and regulations.

"Some data is fine to be shared if it is not identifiable."

(#307, other (health and geography), 1–5 years of experience)

"Only deidentified data is made available upon request.

(#322, nursing, 1–5 years of experience)

"I do share deidentified data in the sense that I publish it"

(#252, health informatics, 16–20 years of experience)

Opening and sharing data is specific to each subject, as expressed by several participants:

"Geographic data can be useful for epidemiologic studies."

(#272, veterinary science, 16–20 years of experience)

Deidentification and anonymization should be done carefully, as several participants expressed concerns that deidentified data can be relinked and identified back to individuals.

"In small data set it's often hard to guarantee anonymity.

(#191, medical science, 21 years and above of experience)

"Small cells that can be used to trace back to individuals. Indirect data may be traced back to individuals if sample sizes are small."

(#201, public health, 16–20 years of experience)

"Even with anonymization that people could be identified"

(#227, health informatics and information technology, 6–10 years of experience)

Any information can be identified back to an individual when the conditions are right.

(#307, other (health and geography), 1–5 years of experience)

"A data science or cybersecurity professional can triangulate data elements related to people and map them to a specific identity. Care should be taken on how much data is made publicly available if a person wants to remain anonymous."

(#346, medical science, 21 years and above of experience)

**Table 6. Health scholars' motivations/driving factors to share health research data.**

| Rank | Motivations to share health research data | Mean | SD |
|---|---|---|---|
| 1 | For transparency of my research. | 4.25 | 1.019 |
| 2 | To increase credibility to research findings. | 4.19 | 1.07 |
| 3 | For knowledge development. | 4.14 | 1.14 |
| 4 | To comply with national mandates and policies of open data. | 3.98 | 1.186 |
| 5 | It was required by the funding agency or sponsor. | 3.65 | 1.309 |
| 6 | To find alternative hypotheses. | 3.59 | 1.201 |
| 7 | It was professionally expected. | 3.46 | 1.271 |
| 8 | To comply with my university/institution policy. | 3.44 | 1.326 |
| 9 | To get acknowledgment as co-authorship. | 2.92 | 1.291 |
| 10 | To get award, career advancement, and recognition. | 2.51 | 1.287 |

## Factors influencing health scholars to share research data

This section examines factors influencing health scholars to share their research data. The study employed a Likert scale survey to collect responses from the participants to rate these factors. The survey consists of questions exploring factors such as the perceived benefits of data sharing, challenges, and potential issues and risks associated with sharing research data.

Table 6 displays the results on motivations of health scholars to open and share their research data. The results show that the top three motivations for health researchers to share their research data are research transparency, increased credibility of research findings, and knowledge development. Respondents rated these motivations highly, with mean scores of 4.25, 4.19, and 4.14, respectively. Compliance with national mandates and policies was also a good motivation to share research data, with a mean score of 3.98. Similar to policy compliance, funding agency or sponsor requirements was also one of the driving factors to share research data, with a mean of 3.65. It was found that the ability to find alternative hypotheses, meet professional expectations, and comply with institutional policy were motivations that health researchers agreed with to a lesser extent, with mean scores of 3.59, 3.46, and 3.44, respectively. Acknowledgment of co-authorship and obtaining awards, career advancement, and recognition were rated even lower, with mean scores of 2.92 and 2.51, respectively.

The findings suggest that health researchers are primarily motivated by transparency, credibility, and knowledge development when sharing their research data. This underscores the importance of providing researchers with the tools and support to make their data available to others to advance scientific discovery and improve health sciences. The result is similar to Zhu's [15] study in that reward was not seen as a strong motivation for sharing data. There were not many rewards resulting from sharing research data.

The study compared motivations according to groups of discipline using Kruskal-Wallis. There are significant differences in disciplines on motivations for sharing research data: for transparency of research ($H = 19.984$, $df = 9$, $p < 0.05$) and to comply with national mandates and policies of open data ($H = 18.044$, $df = 9$, $p < 0.05$), as presented in Table 7.

The mean ranks for the motivation for sharing research data for transparency of research are as follows (ascending): pharmaceutical science (MR = 125.85), healthcare administration (MR = 156.47), nursing (MR = 163.84), medical science (MR = 173.93), psychology (MR = 178.6), health informatics and information technology (MR = 179.48), other (MR = 191.95), public health (MR = 195.48), veterinary science (MR = 232.56), and dentistry (MR = 247.19). Based on pairwise comparisons, pharmaceutical science has significantly lower

**Table 7. Test for group differences across disciplines of health scholars on motivations for sharing research data.**

| Motivations to share health research data | Kruskal-Wallis H | df | Asymp. Sig. |
|---|---|---|---|
| For transparency of my research. | 19.984 | 9 | 0.018 * |
| To increase credibility to research findings. | 12.467 | 9 | 0.188 |
| For knowledge development. | 9.25 | 9 | 0.415 |
| To comply with national mandates and policies of open data. | 18.044 | 9 | 0.035 * |
| It was required by the funding agency or sponsor. | 6.832 | 9 | 0.655 |
| To find alternative hypotheses. | 5.135 | 9 | 0.822 |
| It was professionally expected. | 12.809 | 9 | 0.171 |
| To comply with my university/institution policy. | 11.365 | 9 | 0.251 |
| To get acknowledgment as co-authorship. | 7.695 | 9 | 0.565 |
| To get award, career advancement, and recognition. | 9.249 | 9 | 0.415 |

* $p < 0.05$;

** $p < 0.01$.

motivation for transparency from other disciplines, except healthcare administration and nursing.

In terms of complying with national open data mandates and policies as a motivation to share research data, the disciplines are ranked based on mean ranks (ascending): healthcare administration (MR = 114.57), pharmaceutical science (MR = 130.29), health informatics and information technology (MR = 173.84), nursing (MR = 175.4), medical (MR = 176.87), veterinary science (MR = 185.14), dentistry (MR = 189.88), psychology (MR = 187.98), public health (MR = 204.7), and other disciplines (MR = 212.3). The pairwise comparison shows that healthcare administration is a significantly lower mean rank than other disciplines, except pharmaceutical science and health informatics and information technology.

The Kruskal-Wallis test was also performed to understand the difference in mean rank between grant funding and motivations for sharing research data. The test revealed that there are statistically significant differences between funded scholars on: "to comply with national mandates and policies of open data" (H = 30.957, df = 2, $p < 0.01$), "it was required by the funding agency or sponsor" (H = 38.585, df = 2, $p < 0.01$), and "to get acknowledgment of co-authorship" (H = 6.481, df = 2, $p < 0.05$) as presented in Table 8.

Mean ranks for the motivation to share research data because of "to comply with national open data mandates and policies" are as follows: unfunded/self-funded (MR = 148.08), institutionally funded (MR = 150.64), and nationally funded (MR = 206.62). According to pairwise comparisons, nationally funded health scholars have a higher perception of sharing research data to comply with national mandates and policies. Similarly, the mean rank for nationally funded scholars (MR = 207.32) is also found to be significantly higher regarding "it was required by funding agency or sponsor," compared to institutionally funded (MR = 139.54) and unfunded/self-funded (MR = 146.07).

Regarding the motivation "to get acknowledgment of co-authorship," the mean ranks for funded health scholars are as follows in ascending order: nationally funded (MR = 166.27), institutionally funded (MR = 183.92), and unfunded/self-funded (MR = 198.98). The pairwise comparisons show that nationally funded health scholars have a significantly lower mean rank than unfunded/self-funded ones in terms of seeking acknowledgment of co-authorship when sharing research data.

Several participants expressed the importance of policy and agreements in driving them to share their research data. Many policies in funded research require researchers to share their

**Table 8. Test for group differences across grant funding of health scholars on motivations for sharing research data.**

| Motivations to share health research data | Kruskal-Wallis H | df | Asymp. Sig. |
|---|---|---|---|
| For transparency of my research. | 3.024 | 2 | 0.221 |
| To increase credibility to research findings. | 0.978 | 2 | 0.613 |
| For knowledge development. | 0.499 | 2 | 0.779 |
| To comply with national mandates and policies of open data. | 30.957 | 2 | 0.000 ** |
| It was required by the funding agency or sponsor. | 38.585 | 2 | 0.000 ** |
| To find alternative hypotheses. | 4.887 | 2 | 0.087 |
| It was professionally expected. | 1.784 | 2 | 0.41 |
| To comply with my university/institution policy. | 1.271 | 2 | 0.53 |
| To get acknowledgment as co-authorship. | 6.481 | 2 | 0.039 * |
| To get award, career advancement, and recognition. | 2.505 | 2 | 0.286 |

* $p < 0.05$;

** $p < 0.01$.

research data. These requirements are the incentives that motivate researchers to share their research data.

"If the funder requires it, that's the strongest incentive."

(#154, clinical psychology, 11–15 years of experience)

"Data sharing is a mandatory now trough federal funding."

(#405, public health, 21 years and above of experience)

"We are funded by a foundation that requires sharing."

(#483, public health, 11–15 years of experience)

Several participants commented on the benefits of sharing health research data, such as science and knowledge development, as driving factors. One of the participants commented that it was the right thing to do.

"It needs to be always public for the benefit of the other researchers to progress.

#405

Accelerate learning in the field of study."

(#156, veterinary science, 6–10 years of experience)

"Because it is the right thing to do."

(#342, medical science, 21 years and above of experience)

Health scholars face several challenges and issues that can impede the sharing of their research data, as shown in Table 9. The primary challenge faced by researchers is the concern that their research data contains private and personal information (3.93), followed closely by the concern that the data is confidential (3.73) and the lack of support and resources to share the data (3.23). The results highlight that protecting sensitive data is a crucial factor researchers

**Table 9. Health scholars' perceptions toward challenges and issues to open and share health research data.**

| Rank | Challenges and Issues to Open and Share Health Research Data | Mean | SD |
|---|---|---|---|
| 1 | My research data contains private and personal data. | 3.93 | 1.338 |
| 2 | My research data is confidential. | 3.73 | 1.322 |
| 3 | Lack of support and resources to do it. | 3.23 | 1.363 |
| 4 | I do not have shareable data. | 2.89 | 1.422 |
| 5 | Others can misinterpret and misapply my research data. | 2.88 | 1.298 |
| 6 | There is no clear usage license in my institution. | 2.73 | 1.237 |
| 7 | There is no obligations, requirements, and policies. | 2.71 | 1.245 |
| 8 | There is no career reward. | 2.64 | 1.288 |
| 9 | I do not know how to. | 2.54 | 1.351 |
| 10 | I do not trust others to use my research data. | 2.47 | 1.291 |
| 11 | I want to retain exclusive use of it. | 2.43 | 1.308 |
| 12 | To protect from other competitors. | 2.28 | 1.297 |

consider before sharing their data. The other challenges and issues scored less than 3. The finding means that some participants found that these issues and challenges are happening in some places but are not major, including do not have sharable data (2.89), potential misinterpretation or misapplication of data by others (mean score of 2.88), the absence of clear usage licenses and policies (2.73), no obligation or requirements (2.71), no career rewards (2.64), and do not how to (2.54). The rest of the challenges and issues scored lower than 2.5. The results show that it is rarely an issue in many places, including do not trust others (2.47), wanting to retain an exclusive use (2.43), and protecting from other competitors (2.28). Ethical concerns have been long-running challenges regarding research data sharing [57]. Privacy, consent, and anonymization issues in open research data can be found in many cases.

The study also examined the mean ranks of health scholars across different disciplines in their perceptions of the issues and challenges of sharing research data. The Kruskal-Wallis test revealed significant differences of disciplines in: "I do not have shareable data" (H = 17.9, df = 9, p < 0.05), "I do not how to" (H = 18.013, df = 9, p < 0.05), and "to protect from other competitors" (H = 18.277, df = 9, p < 0.05) as presented in Table 10.

The mean ranks for the challenge of "I do not have sharable data" across disciplines are as follows (ascending): veterinary science (MR = 141.83), health informatics and information technology (MR = 154.22), psychology (MR = 155), dentistry (MR = 158.69), nursing (MR = 162.63), medical (MR = 170.57), healthcare administration (MR = 193.56), pharmaceutical science (MR = 203.16), other (MR = 211.42), and public health (MR = 212.12). The higher score in this category indicates that scholars in specific disciplines do not share data because they do not have sharable data. Through pairwise comparisons, it is found that public health has a significantly higher mean rank compared to veterinary, health informatics and information technology, and nursing.

The mean ranks for the challenge of "I do not know how to" across disciplines are as follows (ascending): dentistry (MR = 120.25), health informatics and information technology (MR = 135.86), psychology (MR = 156.57), public health (MR = 166.28), veterinary science (MR = 173.89), healthcare administration (MR = 179.56), other (MR = 182.39), medical (MR = 185.98), pharmaceutical science (MR = 189.13), and nursing (MR = 217.73). A higher score in this category indicates that scholars in specific disciplines do not know how to share data. Through pairwise comparisons, it is found that scholars in nursing have significantly higher mean ranks than all other disciplines, except the medical science scholars' group, who

**Table 10. Test for group differences across disciplines of health scholars on perceptions of challenges and issues in sharing research data.**

| Challenges and Issues to Share Health Research Data | Kruskal-Wallis H | df | Asymp. Sig. |
|---|---|---|---|
| My research data contains private and personal data. | 8.534 | 9 | 0.481 |
| My research data is confidential. | 12.242 | 9 | 0.2 |
| Lack of support and resources to do it. | 11.832 | 9 | 0.223 |
| I do not have shareable data. | 17.9 | 9 | 0.036 * |
| Others can misinterpret and misapply my research data. | 3.335 | 9 | 0.95 |
| There is no clear usage license in my institution. | 9.597 | 9 | 0.384 |
| There is no obligations, requirements, and policies. | 12.529 | 9 | 0.185 |
| There is no career reward. | 8.863 | 9 | 0.45 |
| I do not know how to. | 18.013 | 9 | 0.035 * |
| I do not trust others to use my research data. | 8.935 | 9 | 0.443 |
| I want to retain exclusive use of it. | 10.152 | 9 | 0.338 |
| To protect from other competitors. | 18.277 | 9 | 0.032 * |

\* $p < 0.05$;

\*\* $p < 0.01$.

followed closely after. Medical science scholars also have significantly higher mean ranks than scholars in health informatics and information technology.

The mean ranks for the challenge of "to protect data from other competitors" across disciplines are as follows: dentistry (MR = 113.31), veterinary science (MR = 147.97), public health (MR = 157.34), health informatics and information technology (MR = 161.92), psychology (MR = 188.67), nursing (MR = 188.94), medical (MR = 174.93), other (MR = 205.61), pharmaceutical science (MR = 221.44), and healthcare administration (MR = 226.34). A higher score in this category indicates that scholars agree that they do not share data to protect their research data from other competitors. Pairwise comparisons revealed that healthcare administration and pharmaceutical science scholars express greater concern about protecting their data from competitors, as evidenced by a significantly higher mean rank than other disciplines.

The Kruskal-Wallis test was also employed to test mean rank differences across funded scholars on their perceptions of the issues and challenges related to sharing research data; results are displayed in Table 11. A statistically significant difference was detected in grant funding towards perceptions of challenges and issues in sharing research data: "lack of support and resources to do it" (H = 8.12, df = 2, $p < 0.05$), "there is no clear usage license in my institution" (H = 7.131, df = 2, $p < 0.05$), and "I do not know how to" (H = 7.172, df = 2, $p < 0.05$). The higher the mean ranking score, the higher concerns for caring for those challenges and issues.

The mean ranks for the challenge of "lack of support and resources to do it" across funding grants are as follows (ascending): institutionally funded (MR = 157.61), nationally funded (MR = 166.18), and unfunded/self-funded (MR = 200.17). A higher mean rank score indicates that scholars in the respective group prefer not to share their research data due to a lack of support and resources. Pairwise comparisons revealed that unfunded/self-funded health scholars significantly differ from funded scholars in support and resources.

The mean ranks for the challenge of "lack of clear usage license in my institution" across funding grants are as follows (ascending): nationally funded (MR = 166.29), institutionally funded (MR = 167.54), and unfunded/self-funded (MR = 199.04). A higher mean rank score

**Table 11. Test for group differences across grant funding of health scholars on perceptions of challenges and issues in sharing research data.**

| Challenges and Issues to Share Health Research Data | Kruskal-Wallis H | df | Asymp. Sig. |
|---|---|---|---|
| My research data contains private and personal data. | 0.641 | 2 | 0.726 |
| My research data is confidential. | 0.506 | 2 | 0.777 |
| Lack of support and resources to do it. | 8.12 | 2 | 0.017 * |
| I do not have shareable data. | 2.358 | 2 | 0.308 |
| Others can misinterpret and misapply my research data. | 1.88 | 2 | 0.391 |
| There is no clear usage license in my institution. | 7.131 | 2 | 0.028 * |
| There is no obligations, requirements, and policies. | 5.976 | 2 | 0.05 |
| There is no career reward. | 0.455 | 2 | 0.797 |
| I do not know how to. | 7.172 | 2 | 0.028 * |
| I do not trust others to use my research data. | 1.168 | 2 | 0.558 |
| I want to retain exclusive use of it. | 4.535 | 2 | 0.104 |
| To protect from other competitors. | 5.217 | 2 | 0.074 |

\* $p < 0.05$;

\*\* $p < 0.01$.

indicates that scholars in the respective group do not share their research data due to the absence of a clear usage license in their institution. Similarly, according to pairwise comparisons, unfunded/self-funded health scholars significantly differ from funded scholars regarding the presence of a clear usage license.

The mean ranks for the challenge of "lack of knowledge on how to do it" across funding grants are as follows (ascending): nationally funded (MR = 166.29), institutionally funded (MR = 167.54), and unfunded/self-funded (MR = 199.04). Similar to the previous significant challenges and issues, according to pairwise comparisons, unfunded/self-funded health scholars significantly express more concern regarding their knowledge of how to share their research data than funded scholars.

Non-funded health science researchers have the highest scores for all significant perceptions toward challenges and issues. This result is understandable since these researchers did not get enough funding for their research activities, including sharing their research data. Since they lacked resources and support, they are also likely to encounter an issue of do not how to, found that there was no obligation nor clear usage license.

The study provides important insights into why researchers do or do not share their research data. The findings suggest that funding agencies and sponsors can play a crucial role in promoting open data by requiring researchers to share their data as a condition of funding. Institutions can also support open data by providing researchers with clear usage licenses and resources, including non-funded ones.

There are many open-ended responses that can be used to make sense of findings of the survey. Many participants expressed concerns regarding the difficulties of deidentification and anonymization. According to several participants, several data types are difficult to deidentify, such as qualitative data.

"Qualitative data relating to illicit behaviors. As it is not possible to reliably deidentify such data"

(#396, public health, 21 years and above of experience)

"Qualitative data which cannot be sufficiently deidentified"

(#477, public health, 11–15 years of experience)

Even with deidentification and anonymization, several participants expressed risks when sharing their data, as commented by participants:

"I generally do NOT share data as it often contains sensitive PHI (e.g. mental health data). Even when every effort is made to deidentify data, it feels risky to me"

(#393, clinical psychology, 21 years and above of experience)

"Sometimes researchers worry about that even with deidentified data."

(#249, other (food and nutrition science), 6–10 years of experience)

Besides the deidentification process, sharing research data requires extra effort to make sure that it supports reuse and is usable for others.

"data must be reformatted with much more explanation so that the others understand the sample, data collection idiosyncrasies, limitations to generalizability, conventions of data formatting and coding by the study, manner in which constructed variables were created, data editing and cleaning procedures"

(#141, medical science, 6–10 years of experience)

"I don't have the time to exhaustively review."

(#209, health informatics and information technology, 11–15 years of experience)

"..data must be reformatted with much more explanation so that the others understand the sample.. has no funds or resources to help"

(#389, medical sciences, more than 20 years of experience)

The extra efforts require additional resources that several participants commented that they were unwilling to share, as mentioned by one of the participants:

"It takes a great amount of effort to procure funding to collect data, and then even more effort to collect the data. We do not give this effort away for free to others who want our data."

(#303, medical science, 21 years and above of experience)

Resources and support can be offered in the form of user education and training. Although the mean value for the issue of "do not know" was low, indicating that it was not a significant concern, some participants expressed the need for training in this area. They believe that the lack of training and preparation in sharing research data can lead to additional issues in data preparation.

"There's not training or funds."

(#339, other, 21 years and above of experience)

"I'll need to be trained on this."

(#324, psychology, 11–15 years of experience)

In the previous part, it is mentioned by several participants that policies and agreements are the incentives that motivate scholars to share their research data. However, there were also comments that agreements can be an inhibiting factor in sharing research data, as mentioned by several scholars:

"Data use agreements that prohibit sharing with the general public or even other researchers"

(#198, medical sciences, 16–20 years of experience).

"Agreement that preclude us from sharing the data publicly or with others"

(#323, public health, 11–15 years of experience)

"Data use agreements that prohibit sharing with the general public"

(#154, clinical psychology, 11–15 years of experience)

"Some projects prohibit truly open data sharing because of institutional policies regarding sensitive clinical data."

(#267, clinical psychology, 6–10 years of experience)

Even though "do not trust others to use my research data" received a low score which means the level of trust is high, one of the participants was concerned that sharing their research data has the potential that others may claim the ownership or may misinterpret the data.

"Plagiarism and recycling of my data by others, who then claim authorship for the same work. Or worse, misinterpret or misrepresent the data and falsely claim that my team falsified findings.

(#251, medical science, 21 years and above of experience)

## Conclusions

This survey study comprehensively investigates the practices and preferences of health scholars regarding the sharing of research data. A survey of 59 questions was distributed and gathered responses from 362 health scholars. Health scholars share their research data due to various factors and depending on different situations. Various factors such as research settings, data types, data formats, grants, journal publications, and institutions contribute to the different preferences for sharing health research data. The findings from the Kruskal-Wallis tests indicate significant differences among disciplines regarding sharing research data. Specifically, these differences were observed in sharing research data in informal ways, conducting secondary data analysis, and opening research data through journal supplements. These results suggest that disciplinary variations exist in the attitudes and behaviors of health scholars toward the sharing of research data. This study added significant findings.

It was also found that there was also a significant difference in grant funding towards the practice of sharing research data in informal ways (personal communication). Health scholars share their research data informally through personal communication, which is more pronounced among unfunded/self-funded research projects. Informal ways sharing of research data is a prevalent practice among researchers, and it is a valuable avenue for promoting open

data practices. These findings can inform policies and practices that promote open data practices in health research, particularly among researchers who lack the necessary support and resources to share their data.

Another notable finding from this study is that compliance with national open data mandates and policies is a significant driving factor for sharing data, especially among nationally-funded research projects. This finding highlights the importance of open data policies and mandates in promoting data sharing in health research. Another interesting finding is that the funding agency or sponsor requirement is another significant motivator for sharing data, which is more pronounced among nationally funded projects. This suggests that funding agencies and sponsors have a critical role in promoting open data practices by making it a requirement for research projects they fund.

Many health scholars are driven to share their research data due to research transparency, increasing research findings' credibility, and knowledge development. This study's results indicate that health scholars are more interested in science and knowledge development, resulting from sharing research data.

There are still challenges and issues in the lack of resources and support, especially for non-funded scholars. There are also difficulties in the process of deidentification and anonymization of specific types of data. The deidentification process should be conducted thoroughly to prevent the reidentification of data, which could potentially link back to individuals. Challenges related to trust, exclusive use, and intellectual property persist in the context of sharing health research data. Lack of support and resources to share data is another factor that influences researchers' decisions to share their data. The mean score is relatively high, indicating that researchers may require more support and resources to share their data effectively. Several participants expressed concerns on the need of training on data preparation and the proper methods for sharing data. The results highlight the importance of providing researchers with the necessary resources, training, and support to facilitate open data practices.

This study empirically examined factors influencing the sharing of research data among health science scholars, expanding upon existing literature. The findings confirmed several prior studies, indicating low utilization of public health research data in university settings and the limited role of rewards as a motivation for data sharing. The findings of this study can help us develop strategies that incentivize researchers' involvement in sharing health research data. These results can inform the development of policies and guidelines for open data and data sharing in the health sciences field.

## Limitations and future research

While this study contributes to the literature by examining how health scholars share their research data, further research is necessary to explore the models and factors influencing their decisions. Currently, this study provides preliminary results toward describing the practices and views of health scholars regarding the sharing of their research data. However, many other factors remain unexplored, including the number and value of winning grants, involvement in collaborative projects, research experience, and other demographic information about survey participants. Future research could focus on investigating these factors. The data resulting from the survey allows for further exploration in this area. Participants have previously expressed that the decision to open and share research data depends on various factors. Addressing these factors will provide a more comprehensive understanding of why health scholars are interested in sharing their research data.

In addition to investigating the decision to share health research data, it is important to examine the extent of data reuse and utilization by others. Sharing research data aims to allow

the data to be reusable by others to increase the credibility of findings and maximize further discoveries. However, this study has identified a relatively low level of data reuse by health science scholars, indicating the need for further exploration in this area. Understanding the patterns and motivations behind data reuse can provide valuable insights into the impact and effectiveness of open data initiatives. Further studies in this area can help us better understand how health scholars are utilizing publicly available open data and identify any barriers or challenges that may exist.

## Author Contributions

**Conceptualization:** Muhamad Prabu Wibowo, Lorri Mon.

**Formal analysis:** Muhamad Prabu Wibowo.

**Methodology:** Muhamad Prabu Wibowo.

**Supervision:** Lorri Mon.

**Writing – original draft:** Muhamad Prabu Wibowo.

**Writing – review & editing:** Lorri Mon.

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
