## [Decision Letter · Decision Letter 0]

3 Jun 2024

PONE-D-24-14271Investigating the practices and preferences of health scholars in opening and sharing research dataPLOS ONE

Dear Dr. Wibowo,

Thank you for submitting your manuscript to PLOS ONE. After careful consideration, we feel that it has merit but does not fully meet PLOS ONE’s publication criteria as it currently stands. Therefore, we invite you to submit a revised version of the manuscript that addresses the points raised during the review process. More specifically, the Reviewers have raised important aspects that must be addressed in a revised version of your manuscript:Reviewer 1 and Reviewer 3 request more clarity regarding the actual meaning “opening” and “sharing” data. I agree that defining these, partially overlapping but different, terms clearly in the beginning of the manuscript will be helpful for readers. In addition, further clarifications regarding whether and how specific practices are really events of data sharing, e.g., sharing within a research team, are required. I think that a clear definition will be helpful to contextualize and interpret these events.Based on the comments and suggestion made by Reviewer 1 and Reviewer 2, I ask you to expand the concluding section and to discuss your main findings in the context of the existing literature (see also Reviewer 1’ s suggestion for restructuring and moving contents from the Findings). Revising the Conclusion may also help you to emphasize what your study adds to the literature compared to previous studies – an issue that has been raised by Reviewer 3.Finally, a thorough language editing will be helpful avoid some not well take phrases and to improve the readability of your manuscript.

We look forward to receiving your revised manuscript.

Kind regards,

Bastian Rake

Academic Editor

PLOS ONE

“The study is funded by the American Indonesian Cultural and Educational Foundation (AICEF) dissertation travel grant and the Florida State University, School of Information, Esther Maglathlin Doctoral Research Scholarship.”

Reviewers' comments:

Reviewer's Responses to Questions

**Comments to the Author**

1. Is the manuscript technically sound, and do the data support the conclusions?

Reviewer #1: Partly

Reviewer #2: Partly

Reviewer #3: Yes

2. Has the statistical analysis been performed appropriately and rigorously? 

Reviewer #1: I Don't Know

Reviewer #2: Yes

Reviewer #3: Yes

3. Have the authors made all data underlying the findings in their manuscript fully available?

Reviewer #1: Yes

Reviewer #2: Yes

Reviewer #3: Yes

4. Is the manuscript presented in an intelligible fashion and written in standard English?

Reviewer #1: Yes

Reviewer #2: Yes

Reviewer #3: Yes

5. Review Comments to the Author

Reviewer #1: This manuscript reports the results of a survey of US-based health science researchers, which seeks to understand respondents’ attitudes and behaviors around research data sharing. It builds on an extensive (and, here, well-synthesized) body of scholarship examining drivers and barriers of data sharing, taking up these questions in the specific context of the health sciences. The items and fixed-choice options developed for the survey instrument are grounded in the existing literature; I do not have the expertise to comment on the statistical analysis undertaken.

It is notable that the manuscript introduces the phrase “opening and sharing health data” in its opening sentences, without going on to define opening or sharing and how they might differ or overlap. This may sound like a cosmetic point, but uncritically coupling these concepts leads to statements like: “Opening and sharing health research data can be done by making available through data repositories, journal supplements, and peer-to-peer” (lines 117-118). Yet data that is shared privately between peers is, presumably, not open data—in the sense of being “data that can be freely used, re-used and redistributed by anyone” (https://opendatahandbook.org/guide/en/what-is-open-data/), generally in the context of an open license and/or public domain dedication.

There are moments in the manuscript where I struggle to understand what the authors mean by open. For instance, they report that “surveyed health scholars prefer to share their data through less formal and smaller scope of data-sharing channels. [These scholars] prefer internal and personal methods of opening and sharing health research data instead of through formal repositories and public platforms” (lines 336-339). I see how this statement follows from the data presented in Table 2 and it is an important finding, which confirms a preference for private sharing found in other researcher surveys (https://doi.org/10.7717/peerj.16731). But would the authors really say that sharing data within a team constitutes opening the data at all? Likewise, they conclude that “informal ways [of] sharing data is [sic] a prevalent practice among researchers, and it is a valuable avenue for promoting open data practices” (lines 770-772). Are the authors suggesting that informal and private sharing may subsequently lead to open sharing and, if so, on what basis?

I am emphasizing this point not to be pedantic about the authors’ use of language, but because research shows that private approaches to data sharing have effects that are different than open approaches. When data are made available upon request, these requests are not infrequently ignored, lost, or subject to undisclosed restrictions (https://doi.org/10.1038/s41597-021-00981-0). The point here is not that private sharing is bad and open sharing is good. But private sharing does carry system-level costs, even as it may also mitigate some of the risks around data sharing that the respondents perceive, and I would like to see some reflection on this tradeoff in the manuscript (as well as a clearer definitional treatment of opening vs. sharing, as early in the paper as possible).

At a more structural level, I would encourage the authors to consider expanding their Conclusions section and moving some of the more interpretive material from the Findings section (e.g, lines 681-690) down into it. The manuscript devotes quite a lot of space to exploring between-group variance; at times, the import of these findings was apparent to me (e.g., funded researchers are more susceptible to funder policies) but at others, especially around disciplinary differences, I found myself wondering “what’s the takeaway here?”

An expanded Conclusions section could also devote more space to unpacking what I see as a very interesting finding, mostly unremarked upon in this version of the manuscript. Table 6 shows that respondents ranked “to get award, career advancement, and recognition” the lowest among possible motivations to share research data. As the authors note on line 528, this may reflect the fact that, descriptively speaking, “there [are] not many rewards resulting from sharing research data.” But then Table 9 shows that respondents ranked “There is no career reward” fairly low among barriers to sharing research data: much lower than professional/ethical considerations around privacy and confidentiality. Open Science advocates often argue that adoption of practices like data sharing has lagged due to the lack of extrinsic reward and recognition (https://doi.org/10.1098/rsos.221460). The results presented here complicate that narrative, and I would very much like to see the authors reflect on what that means.

I look forward to reviewing a revised version of the manuscript that addresses the points above; a language edit would also be helpful to smooth out some occasional awkward phrasing, such as “the U.S. was one of the highest numbers of countries” (lines 245-246).

Reviewer #2: In the method section there are some paragraphs that has not too much sense in this section

The U.S. is recognized as one of the leading countries in providing open data and data repositories, with

numerous government institutions and universities hosting health data repositories. In 2019, the

U.S. was one of the highest numbers of countries that provide data repositories and open data

(60). Many U.S. government institutions host open health data, such as NIH (61,62). It is also

very common for U.S. universities to host data repositories (63). Therefore, the U.S. was selected

as the research setting to capture best practices in open and sharing health research data.

I understand that you want to provide some context, but it coud have more sense in the introduction section or into the lit. review

Which version of SPSS ?

In the section results and finding, the excerpts, do not you have demographic data ? age, gender.. please put also this data if you have it.

For example:Different journals and funders have different requirements for data sharing. I generally

follow the minimum data sharing required” (#141, medical sciences, 5-10 years of

experience)

Table 6. You have means and SD. Maybe can be represented into a graphic instead of a table.

In the Results and discussion section there is no discussion against literature. I suggest you to modify the section and provide comparisons with your study with similar studies and provide a discussion.

Reviewer #3: Thank you for the opportunity to review this paper. The paper investigates the data-sharing practices among health researchers.

General Comments:

The article is well-written and well-structured, making it easy to read. The research question is clear, and the theoretical framework and positioning of the paper are quite well articulated. The methodology is thoroughly described, and the analysis is well executed. I commend the authors for the quality of their work in the first submitted version, which is not often the case. The topic is highly relevant in the context of open science, as well as scientific integrity and the reproducibility of results.

Comments on Content:

- The authors provide a comprehensive literature review, noting that many papers have addressed this question using different contexts, methods, and data. All the research questions posed in this article have already been explored. While the analysis is well-conducted, it is not clear what new insights this study provides compared to the existing literature. Although the authors attempt to address this on page 4 (lines 90-96), their study’s positioning relative to existing results is absent from the discussion and conclusion.

- Distinction between "opening" and "sharing" data: The authors distinguish well between these two concepts; however, this difference is not thoroughly analyzed in the paper, especially regarding the distinction of practices across disciplines. For instance, it would be beneficial to highlight differences in practices between disciplines that are more inclined to share data informally (and the reasons for this) and those that emphasize openness.

- In interpreting the results of the "test for group differences" tables, the authors merely describe the tables without providing deeper explanations of the results. This is evident, for example, in Table 3.

- The authors used Kruskal-Wallis tests, which indicate that at least one sample stochastically dominates another. However, this test does not specify where this stochastic dominance occurs or for how many group pairs it is present. Tests like Dunn's tests are used to analyze specific sample pairs for stochastic dominance. The authors seem to use a manual comparison to analyze the significance of differences. If this is the case, they should justify this choice; otherwise, they should employ an appropriate statistical method (which I recommend).

I would recommend publishing this article in PLOS ONE once the above points have been addressed.

6. PLOS authors have the option to publish the peer review history of their article (what does this mean?). If published, this will include your full peer review and any attached files.

Reviewer #1: **Yes: **Marcel LaFlamme

Reviewer #2: **Yes: **Juan-José Boté-Vericad

Reviewer #3: **Yes: **Abdelghani Maddi

---

## [Author Response · Author response to Decision Letter 0]

17 Jul 2024

Reviewer #1:

Thank you for your detailed feedback and constructive suggestions. We have revised the manuscript to clarify the distinction between "opening" and "sharing" data early on, primarily using the term "sharing" for clarity throughout the manuscript. The Conclusions section has been expanded to discuss the relationship between our study and existing literature. We appreciate your insights on this matter and believe these revisions strengthen the manuscript.

Reviewer #2:

Regarding the representation of data in table 6, we opted to retain the table format to maintain detailed presentation of the results, ensuring clarity and transparency. Graphical representations were considered; however, we found that tables better suit our need to provide comprehensive details without compromising readability. We believe this approach enhances the accessibility and depth of our findings for readers.

Reviewer #3:

Thank you for your positive feedback and recommendations. We have clarified the novelty and contributions of our study compared to existing literature in the discussion section. We have also provided deeper explanations of the statistical analyses, including justifications for our choice of statistical methods or considerations of alternative tests as recommended.

General Response:

Throughout the manuscript, we have addressed minor language issues and improved overall readability. We believe these revisions have strengthened the manuscript in response to your valuable feedback.

Best regards,

Muhamad Prabu Wibowo

---

## [Decision Letter · Decision Letter 1]

2 Sep 2024

PONE-D-24-14271R1Investigating the practices and preferences of health scholars in sharing research dataPLOS ONE

Dear Dr. Wibowo,

Thank you for submitting your manuscript to PLOS ONE. After careful consideration, we feel that it has merit but does not fully meet PLOS ONE’s publication criteria as it currently stands. Therefore, we invite you to submit a revised version of the manuscript that addresses the points raised during the review process.

While Reviewer 2 is satisfied with the revised manuscript, the two other Reviewers raise some issues that must be addressed prior to publication. My understanding is that the required revisions can be implemented in a straightforward way. More specifically:

Reviewer 1 asks for further elaborations and clarifications regarding the distinction between sharing research data and open research data. While you have already implemented some changes, I agree with Reviewer 1 that the use of the word “open” requires a careful re-evaluation.With respect to your empirical analysis, Reviewer 3 asks for a clearer justification of the appropriateness of the Kruskal-Wallis test in the manuscript. Please submit your revised manuscript by Oct 17 2024 11:59PM. If you will need more time than this to complete your revisions, please reply to this message or contact the journal office at plosone@plos.org. Please include the following items when submitting your revised manuscript:A rebuttal letter that responds to each point raised by the academic editor and reviewer(s). You should upload this letter as a separate file labeled 'Response to Reviewers'.A marked-up copy of your manuscript that highlights changes made to the original version. You should upload this as a separate file labeled 'Revised Manuscript with Track Changes'.An unmarked version of your revised paper without tracked changes. You should upload this as a separate file labeled 'Manuscript'.If applicable, we recommend that you deposit your laboratory protocols in protocols.io to enhance the reproducibility of your results. Protocols.io assigns your protocol its own identifier (DOI) so that it can be cited independently in the future. For instructions see: https://journals.plos.org/plosone/s/submission-guidelines#loc-laboratory-protocols. Additionally, PLOS ONE offers an option for publishing peer-reviewed Lab Protocol articles, which describe protocols hosted on protocols.io. Read more information on sharing protocols at https://plos.org/protocols?utm_medium=editorial-email&utm_source=authorletters&utm_campaign=protocols.

We look forward to receiving your revised manuscript.

Kind regards,

Bastian Rake

Academic Editor

PLOS ONE

Journal Requirements:

Reviewers' comments:

Reviewer's Responses to Questions

**Comments to the Author**

1. If the authors have adequately addressed your comments raised in a previous round of review and you feel that this manuscript is now acceptable for publication, you may indicate that here to bypass the “Comments to the Author” section, enter your conflict of interest statement in the “Confidential to Editor” section, and submit your "Accept" recommendation.

Reviewer #1: (No Response)

Reviewer #2: All comments have been addressed

Reviewer #3: (No Response)

2. Is the manuscript technically sound, and do the data support the conclusions?

Reviewer #1: Partly

Reviewer #2: Yes

Reviewer #3: Yes

3. Has the statistical analysis been performed appropriately and rigorously? 

Reviewer #1: I Don't Know

Reviewer #2: Yes

Reviewer #3: N/A

4. Have the authors made all data underlying the findings in their manuscript fully available?

Reviewer #1: Yes

Reviewer #2: Yes

Reviewer #3: No

5. Is the manuscript presented in an intelligible fashion and written in standard English?

Reviewer #1: Yes

Reviewer #2: Yes

Reviewer #3: Yes

6. Review Comments to the Author

Reviewer #1: The distinction between sharing research data and opening/open research data has not been adequately clarified in this revision. Changing the construction "sharing and opening" to simply "sharing" throughout the manuscript is a good start, but the authors have then muddied the waters by referring to "sharing open health sciences data" or "sharing open research data" at numerous points, including in their research questions (lines 97-102).

This construction creates the expectation that all of the data discussed under this category are open, when this is not the case given the examination of peer-to-peer and within-team sharing practices. To my mind, this construction is also confusing because it suggests that the data in question may already have been open before being shared.

I would encourage the authors to take a standard definition of open (data) and then stringently evaluate every usage of the word "open" in the manuscript to see whether it satisfies that definition. I suspect that most of the references to "open" could simply be removed, placing the focus squarely on sharing—whether openly or not.

Reviewer #2: Authors have addressed comments and clear improved the manuscript. Thanks for your effort on improving it.

Reviewer #3: I thank the authors for their response and the revisions made to their article. I carefully reviewed the revised version, and I appreciate the efforts made to improve the clarity of the study's positioning in relation to the existing literature, as well as the distinction between the concepts of "sharing" and "opening" data.

However, regarding the justification for the use of the Kruskal-Wallis test and the manual comparison of groups, I remain perplexed by the response provided by the authors. In their response, the authors indicated that they had "provided deeper explanations of the statistical analyses, including justifications for the choice of statistical methods or considerations of alternative tests." However, after reviewing the revised version with tracked changes as well as the resubmitted version, I did not find any explicit justification for the exclusive use of the Kruskal-Wallis test followed by manual comparisons for analyzing group differences.

It is important to justify this methodological choice or consider appropriate post-hoc tests, such as Dunn's test, to precisely identify which groups have significant differences. Currently, the absence of such justifications or an alternative methodology leaves a gap in the statistical interpretation of the results.

I therefore recommend that the authors revisit this part of their analysis and include a clear justification for the use of manual comparisons following the Kruskal-Wallis test, or opt for a more appropriate statistical method for multiple group comparisons. I would recommend this article for publication once this issue is resolved.

7. PLOS authors have the option to publish the peer review history of their article (what does this mean?). If published, this will include your full peer review and any attached files.

Reviewer #1: **Yes: **Marcel LaFlamme

Reviewer #2: **Yes: **Juan-José Boté-Vericad

Reviewer #3: **Yes: **Abdelghani Maddi

---

## [Author Response · Author response to Decision Letter 1]

22 Oct 2024

We are grateful for the detailed and constructive feedback provided by you and the reviewers. We have carefully addressed all the comments and made the necessary revisions to improve the manuscript.

The major changes made to the manuscript are as follows:

Reviewer 1 requests re-evaluation of the word “open”. Regarding this request:

We appreciate Reviewer 1’s suggestion regarding the reevaluation of the term “open.” The meanings of sharing and “open” are related. We have re-evaluated the use of the word “open” throughout the manuscript to ensure clarity and precision. However, we have retained the term in certain instances. The distinction between "open" and "share" lies in the degree of accessibility. While sharing can be restricted, open research data refers to making data publicly available while adhering to specific guidelines.

This paper evaluates both data sharing and open research data. Therefore, the use of the term “open” helps explain how these concepts are related. This paper is heavily influenced by Zuiderwijk et al. (2020), in What drives and inhibits researchers to share and use open research data? A systematic literature review to analyze factors influencing open research data adoption (PLoS ONE, Vol. 15, pp. 1–50), where both terms are used to explain the relationship between data sharing and open data. In response, we have added further explanation in the literature review, expanding on the relationship between data sharing and open research data (lines 107–127).

By retaining the term "open" in certain instances, we hope readers will better understand how these concepts are related.

Reviewer 3 requests Justification for the Kruskal-Wallis Test. Regarding this request:

We have also addressed Reviewer 3’s request for further justification of the Kruskal-Wallis test in the methods part (line 277 – 281). We have added references to support the appropriateness of the Kruskal-Wallis test for our data, particularly in cases where the data do not meet the assumptions of parametric tests.

We believe that these revisions have significantly enhanced the quality of our manuscript and have addressed all the concerns raised by the reviewers. We are confident that the revised manuscript is now suitable for publication in PlosOne Journal.

Thank you for considering our revised manuscript. We look forward to your favorable response. Please let us know if there are any further revisions or additional information required.

Sincerely,

Muhamad Prabu Wibowo

On behalf of Lorri Mon

---

## [Editor Report · Decision Letter 2]

29 Oct 2024

Investigating the practices and preferences of health scholars in sharing open research data

PONE-D-24-14271R2

Dear Dr. Wibowo,

We’re pleased to inform you that your manuscript has been judged scientifically suitable for publication and will be formally accepted for publication once it meets all outstanding technical requirements.

Within the production process you may review the following two sentences as there may be an issue around grammar or wording: 

page 6, line 158ff., "a survey study to factors ..."page 7, line 182ff., "openness is often avoided, such as in the military..."

Kind regards,

Bastian Rake

Academic Editor

PLOS ONE
---

## [Editor Report · Acceptance letter]

20 Nov 2024

PONE-D-24-14271R2 

PLOS ONE

Dear Dr. Wibowo, 

I'm pleased to inform you that your manuscript has been deemed suitable for publication in PLOS ONE. Congratulations! Your manuscript is now being handed over to our production team.

Kind regards, 

on behalf of

Dr. Bastian Rake 

Academic Editor

PLOS ONE